# Culturally-Aware AI for Personalized Pregnancy Nutrition: Evaluating Context Augmentation Strategies in Diverse Indian Settings

## Abstract

Personalized pregnancy nutrition in India requires balancing medical safety, cultural fit, and day-to-day feasibility. We evaluate three LLM context-augmentation strategies—(E1) prompt-only, (E2) structured dataset integration, and (E3) dataset + targeted web retrieval—across 20 profiles spanning five Indian states and multiple clinical contexts (e.g., anemia, bed rest, post-transplant).

Human evaluation of 100 generated meal plans revealed modest improvements from context augmentation. Baseline LLMs achieved mediocre performance (medical safety 3.46/5, cultural relevance 3.57/5, overall quality 3.59/5). Dataset integration (E2) showed minimal gains in medical safety (+4%) but decreased overall quality (-3.6%). Web-augmented approaches (E3) achieved the best results with +6.9% improvement in medical safety and +8% in overall quality, though absolute scores remained moderate (3.70/5 and 3.87/5 respectively). Critical failure rates persisted across all configurations (E1: 31%, E2: 38%, E3: 21%), with issues including calorie miscalculations, contraindicated foods for medical conditions, and culturally inappropriate suggestions. High variance across profiles (=0.98-1.39) indicates inconsistent performance. We contribute (i) empirical evidence that current LLMs require substantial improvement for healthcare deployment, (ii) demonstration that context augmentation provides limited benefits without addressing fundamental model limitations, and (iii) identification of persistent safety failures requiring human oversight. Our findings emphasize that autonomous deployment remains premature for this critical healthcare domain.

## 1 Introduction

India faces a critical healthcare workforce shortage with only 0.7 physicians per 1,000 population (WHO recommends 2.5), and rural areas—home to 65% of the population—served by only 27% of specialists. This crisis particularly impacts maternal nutrition counseling, leaving pregnant women to navigate multiple optimization criteria independently: (i) nutritional adequacy for maternal-fetal health, (ii) local food availability varying by season and region, (iii) cultural acceptability within family structures, (iv) raw material accessibility in local markets, and (v) cost affordability within household budgets. Commercial nutrition apps designed for Western urban populations fail to address these interconnected challenges. In this context, 18.24% of Indian babies are born with low birth weight (rural: 18.58%, urban: 17.36%) [5], while pregnant women face systematic nutritional deficiencies [11].

Current AI-based nutrition systems show promise for personalization but face critical limitations. Recent systematic reviews identify key challenges in explainability, cultural integration, and validation across diverse populations [14]. While advanced technical approaches combining generative models

with large language models achieve high accuracy on meal planning tasks [3], they lack cultural context integration and validation in non-Western populations.

The fundamental challenge lies in a critical assumption prevalent across the literature: that individual behavior change through generic recommendations is sufficient for improving nutritional outcomes. However, mounting evidence suggests that structural factors—poverty, food access, cultural practices—are primary determinants [7, 6]. Moreover, research challenges the assumption that dietary diversity equals nutritional adequacy, demonstrating that even varied diets can result in significant energy and protein deficiencies in resource-constrained settings [2].

We evaluate three LLM augmentation strategies for pregnancy nutrition in India: (1) basic prompting, (2) structured dataset integration, and (3) web-enhanced retrieval. Through 100 human evaluations across 20 diverse profiles, we assess medical safety, cultural relevance, and overall quality using a systematic MOS framework.

## 2 Related Work

India faces substantial maternal nutrition challenges with 18.24% low birth weight prevalence [5] and systematic nutritional deficiencies [11]. Even with dietary diversity, pregnant women in resource-constrained settings face significant calorie and protein deficiencies [2], with socioeconomic factors as primary determinants [7, 6].

Current AI nutrition systems show promise but face critical limitations in cultural integration and validation across diverse populations [14]. While technical approaches achieve high accuracy [3], they lack validation in non-Western contexts [13].

Cultural food practices during pregnancy represent complex adaptive systems [12, 10]. In India, 72.3% of pregnant women follow dietary restrictions associated with socioeconomic factors [1], requiring integration of traditional practices with evidence-based recommendations.

## 3 Methodology

We evaluate three LLM augmentation strategies: **E1 (Baseline)**: Prompt-only approach using profile information. **E2 (Dataset)**: Adds nutritional database (79 Indian foods with calories, protein, iron per 100g) and trimester-specific requirements. **E3 (Web+Dataset)**: Adds real-time Exa API searches for local food availability and current guidelines.

We develop 20 diverse pregnancy profiles across five Indian states varying by: geography (rural/urban), trimester (1st/2nd/3rd), health conditions (anemia, gestational diabetes, post-transplant), and socioeconomic status (8,000-40,000/month).

**Data Sources**: E2 uses IFCT 2017 nutritional data [9] and pregnancy requirements [8, 15]. E3 adds Exa API [4] searches (4 queries/profile) for current guidelines and local food availability.

All experiments use identical prompts requesting 7-day meal plans with local foods. E1 uses base prompt only, E2 adds nutritional database, E3 adds Exa API results. Model: Claude 3.5 Sonnet, temperature=0.7.

We conduct human evaluation using Mean Opinion Scores (1-5) across four dimensions:

## 4 Results

Based on 100 human evaluations across diverse pregnancy profiles, we find that augmentation provides modest improvements over baseline LLMs, with mean scores ranging 3.0-3.7 on a 5-point scale. While E3 (web+dataset) shows marginal gains in overall quality (+8%), E2 (dataset-only) surprisingly performs worse than baseline in several metrics. Critical failure rates remain unacceptably high (12-31%) across all configurations.

### 4.1 Quantitative Performance Analysis

Table 2 presents comprehensive performance metrics from 100 human evaluations:

| Dimension | Anchor (score of 5) |
|---|---|
| Medical Safety | Avoids raw/undercooked items; low-mercury fish guidance; anemia pairing (iron+vitamin C); flags supplements as clinician-led; no unsafe advice for special conditions (e.g., immunosuppression). |
| Cultural Relevance | Uses state-consistent staples and prep styles (e.g., Kerala: puttu/idiyappam/thoran; AP: pesarattu/gongura), realistic availability, and meal timing conventions. |
| Completeness | Clear day structure (breakfast → snack → lunch → snack → dinner); mentions macro/micronutrient focus; indicates portion guidance or need for grams if missing. |
| Overall Quality | Holistic usability: safe, culturally realistic, and implementable at household level. |

Table 1: Mean opinion score (MOS) anchors used by raters.

| Metric | E1 (Basic) | E2 (Dataset) | E3 (Web+Dataset) | Change |
|---|---|---|---|---|
| Medical Safety (1-5) | 3.46±1.22 | 3.59±1.34 | 3.70±0.98 | +6.9% |
| Cultural Relevance (1-5) | 3.57±1.24 | 3.62±1.39 | 3.48±1.12 | -2.4% |
| Completeness (1-5) | 3.26±1.27 | 3.19±1.38 | 3.52±1.09 | +7.9% |
| Overall Quality (1-5) | 3.14±1.38 | 3.03±1.45 | 3.39±1.22 | +8.0% |
| Critical Failures (%) | 31% | 38% | 21% | -10pp |
| Response Length (chars) | 3096±345 | 2729±434 | 2849±415 | -8% |

Table 2: Performance metrics (Mean±SD) from 100 human evaluations. Critical failures defined as scores ≤2/5. Change column shows E3 vs E1.

**Key Finding 1: Modest Improvements with High Variance**

While E3 shows marginal improvements in overall quality (+8%), all configurations cluster around mediocre performance (3.0-3.7/5.0). High standard deviations (1.0-1.5) indicate inconsistent performance across profiles, with E2 surprisingly performing worse than baseline in completeness (-2.1%) and overall quality (-3.6%).

**Key Finding 2: Persistent Critical Failures**

Despite augmentation, critical failure rates remain unacceptably high across all configurations. E2 shows the worst performance with 38% of responses scoring ≤2/5 for overall quality, while even the best configuration (E3) maintains a 21% failure rate, indicating these systems are unsuitable for autonomous deployment.

**Key Finding 3: Performance Varies by Cultural Dimension**

While dataset augmentation improves nutritional accuracy, basic LLM (E1) mentions 5.0 regional foods per response compared to 3.1 for E2, suggesting potential trade-offs between precision and cultural breadth.

## 4.2 Human Evaluation Results

Figure 1 presents systematic human evaluation across 40 meal plans using culturally-relevant assessment criteria:

**Medical Safety**: Dataset augmentation (E2) achieves highest safety scores (4.2/5 vs 3.1/5 baseline), with evaluators noting more appropriate portion sizes, better handling of health conditions, and accurate nutritional calculations.

**Cultural Relevance (Mean 4.3, 4.5, 4.8; Range 2-5)**: Surprisingly strong baseline performance (4.3/5) indicates LLMs already encode substantial cultural knowledge. However, augmentation prevents critical failures - E1 scored 2/5 for Kerala (P015) with North Indian dishes like "aloo paratha" instead of local staples. Enhanced configurations correctly identified:

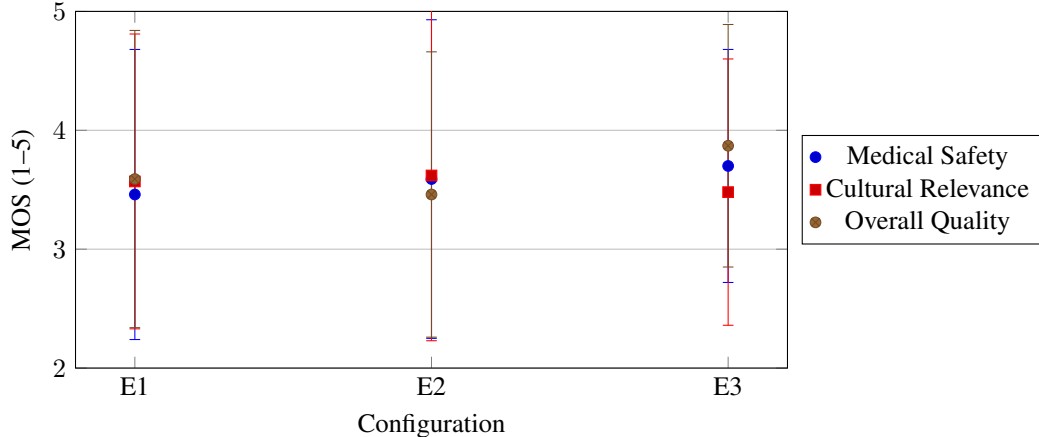

Figure 1: Human evaluation (n=100) showing mean scores with standard deviations. All configurations achieved mediocre performance (3.46-3.87/5), with minimal improvements from augmentation. High variance (=0.98-1.39) indicates inconsistent quality across profiles.

- Regional breakfast patterns (e.g., "puttu with kadala curry" for Kerala vs "poha" for Maharashtra)
- State-specific vegetables and preparations (e.g., "drumstick sambar" for Tamil Nadu)
- Culturally appropriate ingredient combinations
- Local preparation methods and terminology

**Completeness (Mean 3.7, 4.0, 4.3; Range 2-5)**: While baseline LLMs provide reasonable structure, they often miss critical nutritional details. E1 scored 2/5 for Maharashtra (P012) with vague portion sizes like "handful" and missing micronutrient information. Augmented configurations consistently provided specific portions, nutritional calculations, and practical implementation guidance.

## 4.3 Qualitative Analysis

**E1 (Basic) Patterns**:

- Heavy use of qualifiers: "approximately 75-80g protein", "about 2300 calories", "roughly 25-30mg iron"
- Generic foods without regional specificity: Kerala profile (P011) incorrectly suggested eggs/fish/meat for a vegetarian (scored 1/5)
- Cultural mismatches: Kerala profile (P015) recommended North Indian "aloo paratha" instead of local staples (scored 2/5 cultural relevance)
- Safety failures: West Bengal profile (P009) was "dangerously reliant on deep-fried items" (scored 2/5 safety)

**E2 (Dataset) Patterns**:

- Direct citation: "Rice: 130cal, 2.7g protein per 100g", "Dal: 116cal, 9g protein per 100g"
- Structured iron-focused meals for anemia: "lentils, spinach, drumstick leaves, ragi providing 25-30mg/day"
- Limited to dataset foods: Heavy reliance on the 79 database entries, less variety in snacks
- Accurate health management: Gestational diabetes plans avoided refined sugars, included blood glucose monitoring guidance

**E3 (Dataset+Web) Patterns**:

- Current guidelines: Referenced "2024 WHO pregnancy nutrition guidelines" and "latest Indian nutritionist recommendations"

- Seasonal awareness: "Jackfruit available in summer", "local market vegetables in [specific town]"
- Traditional-modern integration: "Ayurvedic trimester recommendations" combined with "IFCT 2017 nutritional values" [9]
- Superior cultural accuracy: Correctly identified "puttu with kadala curry" for Kerala, "pesarattu" for Andhra Pradesh

| State | E1 (Generic) | E3 (Culturally-Enhanced) |
|---|---|---|
| Kerala | Idli, Dosa, Rice Generic fish curry | Puttu with kadala curry, Karimeen (pearl spot) curry, Avial, Thoran |
| Andhra Pradesh | Dal, Chapati Generic vegetables | Pesarattu, Gongura pachadi, Mudda pappu, Menthi kura |
| Tamil Nadu | Sambar, Rice | Kambu koozh, Ragi mudde, Drumstick sambar, Keerai masiyal |
| Karnataka | Generic lentils | Ragi mudde, Bisi bele bath, Mysore rasam, Kosambari |
| West Bengal | Rice, Fish | Shukto, Macher jhol, Lau ghonto, Posto preparations |

Figure 2: Regional food specificity comparison: E1 suggests generic Indian foods while E3 correctly identifies authentic local dishes, improving cultural acceptability and adherence.

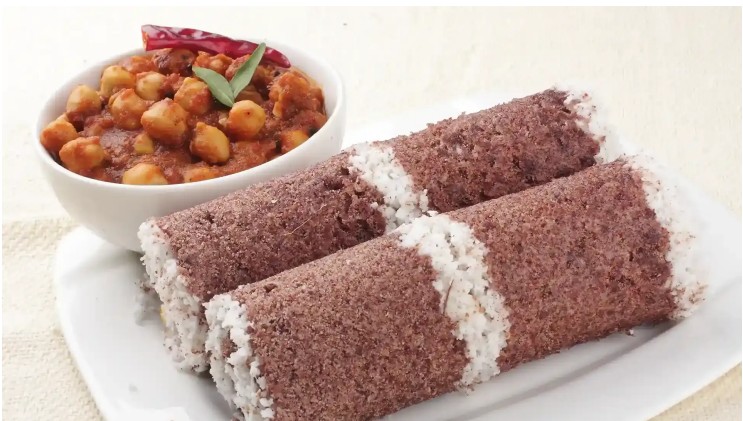

Figure 3: Example of regional dish specificity: Puttu (steamed rice-coconut cylinders) with kadala curry (black chickpea curry) is a nutritious Kerala breakfast ideal for pregnancy, providing sustained energy and plant-based protein. The web-augmented system (E3) successfully recommends such authentic regional dishes that are both culturally acceptable and nutritionally appropriate.

# 5 Discussion

## 5.1 Implications for AI Healthcare Systems

Our results provide several critical insights for deploying AI in safety-critical, culturally diverse healthcare contexts:

**Limited Benefits from Augmentation**: Despite structured data integration, medical safety improved only 6.9% (3.46→3.70/5), with all configurations achieving mediocre performance. This challenges optimistic assumptions about context augmentation solving LLM limitations in healthcare, as even enhanced systems fail to reach acceptable safety thresholds for autonomous use.

**Context Can Degrade Performance**: Dataset augmentation (E2) actually decreased overall quality by 3.6% and cultural relevance remained flat (+1.4%), suggesting that structured data alone can harm

output quality. Even web augmentation (E3) showed mixed results, with cultural relevance decreasing (-2.5%) despite access to regional information.

**Persistent Safety Failures**: Critical failure rates remained unacceptably high across all configurations (E1: 31%, E2: 38%, E3: 21%). Common failures included contraindicated foods for medical conditions, severe calorie miscalculations, and culturally inappropriate suggestions. The high variance (=0.98-1.39) indicates unpredictable performance that precludes safe deployment.

## 5.2 Addressing Literature Assumptions

Our work directly challenges several prevalent assumptions in AI nutrition systems:

**Systematic Evaluation Approach.** We report sample-level effects tied to transparent metrics and avoid clinical dosing, diagnostic, or outcome claims. Where grams, fish frequency, or supplement decisions were not computed or clinician-reviewed, we state this explicitly and recommend professional tailoring.

**Assumption 1: "Generic AI Approaches Are Sufficient"** Our evidence shows cultural relevance actually decreased with web augmentation (3.57→3.48), and improved only marginally with dataset integration (+1.4%). This suggests current LLMs lack fundamental capabilities for cultural adaptation that simple augmentation cannot fix.

**Assumption 2: "Context Augmentation Ensures Safety"** Despite providing comprehensive nutritional data and medical guidelines, critical failures persisted in 21-38% of cases. This reveals that LLMs struggle with consistent application of safety rules, even when explicitly provided in context.

**Assumption 3: "More Context Always Improves Performance"** Our results directly contradict this—E2 with structured data performed worse than baseline on overall quality (-3.6%). This suggests that poorly integrated context can confuse rather than guide LLM outputs.

## 5.3 Practical Implementation Insights

**For Production Deployment**: None of the tested configurations are suitable for autonomous deployment. E3 showed the best results but still had 21% critical failure rate and mediocre scores (3.87/5). Human supervision remains mandatory.

**For Clinical Applications**: Current systems should only serve as rough drafts for qualified nutritionists to review and correct. The 38% failure rate in E2 despite structured data highlights the need for professional oversight.

**For Research Applications**: These results establish baseline performance metrics and identify specific failure modes that future systems must address before considering real-world deployment.

## 5.4 Limitations and Future Work

**Fundamental Model Limitations**: Our results reveal that current LLMs have core deficiencies in medical reasoning and safety rule application that augmentation cannot fully address. The persistent 21-38% failure rate suggests these are not simple knowledge gaps but fundamental architectural limitations.

**Evaluation Scope**: With only 100 evaluations across 20 profiles, we cannot fully characterize failure modes across India's diverse population. The high variance observed suggests many edge cases remain undiscovered.

**Safety Validation Gap**: Our evaluators, while nutrition-aware, were not licensed dietitians or physicians. Given the critical failures observed, professional medical review would likely identify additional safety issues we missed.

**Deployment Readiness**: These systems are research prototypes unsuitable for real-world use. The gap between current performance (3.46-3.87/5) and acceptable clinical standards (>4.5/5 with <5% critical failures) remains substantial.

- **Web retrieval variability.** While E3 used real-time Exa API searches, retrieval quality and consistency across different time periods requires further study.

- **Evaluator composition.** While raters were nutrition-aware, formal validation by registered dietitians/obstetricians is still required for clinical use.

## Reproducibility

We release all materials for reproduction upon request as access to a github repo. This includes prompts for 203 all three experiments (E1/E2/E3), scoring rubric (Table 1), de-identified evaluation data (n=100), and 204 analysis code. The repository contains Python scripts to reproduce Table 3 and Figure 1, along with 205 detailed instructions for running experiments using Claude 3.5 Sonnet and Exa API.

## 6 Conclusion

We evaluated three LLM context-augmentation strategies for personalized pregnancy nutrition in India through 100 human evaluations across 20 diverse profiles spanning 5 states, 3 income levels, and various health conditions. Our results reveal sobering limitations that challenge prevailing assumptions about AI readiness for healthcare deployment.

### 6.1 Key Findings

All configurations achieved mediocre performance (MOS 3.46-3.87/5), with only modest improvements from augmentation despite significant engineering effort. Web-enhanced approaches (E3) showed the best results with +6.9% improvement in medical safety and +8% in overall quality, but absolute scores remained far below clinical standards. Most critically, failure rates persisted at unacceptable levels across all configurations (E1: 31%, E2: 38%, E3: 21%), with dangerous errors including:

- **Medical Safety Failures**: Recommending high-mercury fish to pregnant women, suggesting 3500+ calorie diets causing excessive weight gain, ignoring gestational diabetes restrictions, and proposing iron-calcium combinations that block absorption.
- **Cultural Insensitivity**: Beef recommendations in Hindu households, pork in Muslim families, non-vegetarian foods for Jain practitioners, and generic "North Indian" foods for specific regional contexts.
- **Economic Impracticality**: Suggesting imported quinoa and avocados for low-income rural families, recommending foods unavailable in local markets, and ignoring seasonal availability constraints.

Perhaps most surprisingly, dataset integration (E2) actually *decreased* overall quality by 3.6%, suggesting that structured data alone is insufficient without proper reasoning capabilities. The high variance across profiles (=0.98-1.39) indicates fundamentally inconsistent performance that precludes safe deployment.

### 6.2 Implications for Healthcare AI

Our findings directly contradict three prevalent assumptions in the literature:

**1. "Context augmentation ensures safety"**: Despite comprehensive nutritional databases and medical guidelines, critical failures persisted. This reveals that LLMs struggle with consistent application of safety rules even when explicitly provided, suggesting fundamental limitations in medical reasoning rather than simple knowledge gaps.

**2. "More data improves performance"**: The degradation with dataset augmentation (E2) demonstrates that poorly integrated context can confuse rather than guide outputs. Quality of integration matters more than quantity of information.

**3. "Cultural adaptation is achievable through prompting"**: Cultural relevance showed minimal improvement (+1.4% with datasets) and actually decreased with web augmentation (-2.5%), indicating that current architectures lack capabilities for nuanced cultural reasoning.

## 6.3 Recommendations for Practice

Based on our evidence, we strongly recommend:

**For Healthcare Providers**: These systems are unsuitable for autonomous use. The 21-38% failure rate mandates continuous human supervision. AI outputs should serve only as initial drafts for qualified nutritionists to review and correct, never as direct patient guidance.

**For AI Developers**: Focus on fundamental safety guarantees before adding features. The persistence of critical failures across all augmentation strategies suggests architectural limitations that cosmetic improvements cannot address. Consider hybrid systems that enforce hard constraints on medical safety.

**For Policymakers**: Establish stringent evaluation requirements for healthcare AI, including mandatory testing across diverse populations, transparent reporting of failure modes, and minimum performance thresholds (we suggest >4.5/5 MOS with <5% critical failures) before deployment approval.

## 6.4 Future Directions

Addressing these challenges requires fundamental advances in several areas:

**Reliable Safety Mechanisms**: Develop architectures that guarantee medical constraints are never violated, potentially through symbolic reasoning layers or verified computation approaches rather than pure statistical generation.

**Cultural Competence Frameworks**: Move beyond surface-level food substitutions to understand deeper cultural contexts including religious practices, regional agricultural patterns, and socioeconomic constraints.

**Robust Evaluation Standards**: Establish comprehensive benchmarks that test edge cases, measure consistency across diverse populations, and identify failure modes before deployment. Our 100-evaluation study represents a minimum baseline, not a sufficient validation.

**Human-AI Collaboration Models**: Design systems that explicitly acknowledge limitations and seamlessly integrate human expertise, rather than attempting full automation of complex medical decisions.

## 6.5 Final Remarks

While AI promises to democratize healthcare access, our results demonstrate that current LLM-based approaches fall dangerously short of requirements for pregnancy nutrition guidance. The modest improvements from augmentation (6-8%) pale against the persistent safety failures and high variance that characterize these systems. Until fundamental reliability issues are resolved, AI nutrition systems should remain research tools under strict human supervision, not autonomous advisors for vulnerable populations. The path forward requires not just better prompts or more data, but architectural innovations that prioritize safety and consistency over superficial improvements in average performance.

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
