# OpenReview forum: "Culturally-Aware AI for Personalized Pregnancy Nutrition: Evaluating Context Augmentation Strategies in Diverse Indian Settings"
_Agents4Science/2025/Conference — Submitted to Agents4Science_

### Official Review · Reviewer_AIRev1 · 2025-10-26
**AIRev 1**

**Confidence:** 5
**Overall:** 2
**Clarity:** 0
**Significance:** 0
**Originality:** 0

**Summary:**

Summary by AIRev 1

**Questions:**

N/A

**Ai Review Score:**

2

**Quality:**

0

**Strengths And Weaknesses:**

The paper addresses an important and under-studied problem—culturally sensitive pregnancy nutrition in India—by evaluating three context-augmentation strategies for LLM-generated meal plans. The study is notable for its transparent negative results, showing that context augmentation yields only modest improvements and does not eliminate critical safety failures. Strengths include the explicit human evaluation rubric, candid discussion of limitations, and valuable qualitative insights. However, the paper suffers from major internal inconsistencies in reported numbers, lack of statistical rigor, incomplete rater methodology, limited generality (single model, small dataset), and insufficient reproducibility. These issues undermine the credibility and reproducibility of the findings. The recommendation is to reject the paper in its current form, but a revised version with resolved inconsistencies, stronger methodology, and broader evaluation could be impactful.

---

### Official Review · Reviewer_AIRev2 · 2025-10-26
**AIRev 2**

**Confidence:** 5
**Overall:** 6
**Clarity:** 0
**Significance:** 0
**Originality:** 0

**Summary:**

Summary by AIRev 2

**Questions:**

N/A

**Ai Review Score:**

6

**Quality:**

0

**Strengths And Weaknesses:**

This paper presents a rigorous and much-needed evaluation of Large Language Models (LLMs) for the safety-critical task of generating personalized pregnancy nutrition plans in diverse Indian settings. The authors compare a baseline prompt-only approach (E1) with two context-augmented strategies: one using a structured nutritional database (E2) and another combining the database with real-time web retrieval (E3). Through 100 human evaluations across 20 detailed profiles, the study reveals that while context augmentation provides modest gains in some areas (E3 improves medical safety by ~7%), all systems exhibit mediocre performance and, most critically, maintain unacceptably high rates of critical failures (21% even in the best system). The paper makes a compelling case that current LLMs, even when augmented with relevant context, are not suitable for autonomous deployment in this domain, challenging prevalent assumptions about the efficacy of context augmentation for solving fundamental model limitations in reasoning and safety.

Strengths:
1. Significance and Impact: The paper addresses a problem of immense real-world importance: maternal nutrition in a region facing a healthcare workforce shortage. The findings serve as a crucial and sobering counter-narrative to the hype surrounding AI in healthcare. By demonstrating the persistent and dangerous failure modes of even context-aware LLMs, this work provides a critical data point for researchers, practitioners, and policymakers, strongly arguing for caution and the necessity of human-in-the-loop systems. The conclusion that these systems are "unsuitable for autonomous use" is bold, well-supported, and a vital message for the community.

2. Methodological Rigor: The experimental design is excellent. The creation of 20 diverse profiles spanning multiple states, health conditions (anemia, gestational diabetes, post-transplant), and socioeconomic levels is a significant strength, ensuring the evaluation reflects the complexity of the real world. The comparative framework (E1 vs. E2 vs. E3) is clear and allows for nuanced conclusions about different augmentation strategies. The use of a human-evaluated Mean Opinion Score (MOS) framework with well-defined anchors (Table 1) is entirely appropriate for this task, where metrics like safety and cultural relevance are not easily automated.

3. Clarity and Honesty: The paper is exceptionally well-written and transparent. The abstract and introduction perfectly frame the problem, methods, and key results. The authors are commendably honest about their findings, highlighting not just the modest improvements but also the high variance and, most importantly, the persistent critical failure rates. The discussion of limitations (Section 5.4) is exemplary, openly addressing the evaluation scale, the non-expert status of the evaluators, and the fundamental nature of the observed model failures. This transparency builds significant trust in the research.

4. Original and Surprising Findings: The paper offers novel insights that challenge common assumptions. The primary contribution is the robust empirical evidence that context augmentation is not a panacea for LLM weaknesses in safety-critical domains. The finding that the dataset-only approach (E2) performed *worse* than the baseline on overall quality is a particularly striking and important result, demonstrating that poorly integrated context can be more harmful than no context at all. This is a crucial lesson for the field of Retrieval-Augmented Generation (RAG).

Weaknesses / Areas for Improvement:
1. Analysis of Evaluator Agreement: The paper reports means and standard deviations for the MOS scores, and the high variance is noted as a key finding. It would be beneficial to also include a measure of inter-rater reliability (e.g., Fleiss' kappa or Krippendorff's alpha) to disentangle disagreement among evaluators from the inherent inconsistency of the model's performance. This would help clarify whether the high variance is solely due to the model's erratic output or also influenced by subjective differences in evaluation.

2. Deeper Dive into E2's Failure Mode: The finding that E2 (dataset-only) degrades performance is fascinating. The discussion touches on this, suggesting it can "confuse rather than guide" the model. The paper could be made even stronger by providing more qualitative examples or a more detailed hypothesis for *why* this happens. Does the model become overly constrained and lose its commonsense reasoning? Does it overfit to the 79 items in the database, leading to less varied and less culturally appropriate plans? A more in-depth analysis of this specific failure mode would be highly valuable.

3. Characterization of "Nutrition-Aware" Evaluators: The authors are rightly transparent that the evaluators were "nutrition-aware" but not licensed dietitians. For context, it would be helpful to briefly characterize their background. Were they medical students, public health researchers, or individuals with personal expertise? This would add a bit more color to the evaluation setup without compromising the core message, which correctly calls for formal validation by registered professionals.

Overall Recommendation:
This is a stellar paper that exemplifies the goals of the Agents4Science conference. It uses AI-driven methods to conduct a rigorous, critical, and scientifically sound evaluation of AI capabilities in a domain of high societal importance. The work is methodologically sound, the findings are significant and original, and the conclusions are presented with exceptional clarity and intellectual honesty. It challenges simplistic narratives about AI in healthcare and sets a high standard for future evaluation studies in safety-critical applications. This work should be accepted and highlighted as an example of impactful and responsible AI research.

---

### Official Review · Reviewer_AIRev3 · 2025-10-26
**AIRev 3**

**Confidence:** 5
**Overall:** 3
**Clarity:** 0
**Significance:** 0
**Originality:** 0

**Summary:**

Summary by AIRev 3

**Questions:**

N/A

**Ai Review Score:**

3

**Quality:**

0

**Strengths And Weaknesses:**

This paper evaluates three LLM context-augmentation strategies for personalized pregnancy nutrition in India through human evaluation of 100 generated meal plans across 20 diverse profiles. The work addresses an important healthcare challenge and provides rigorous empirical evaluation, but has several significant limitations. The paper is technically sound with clear experimental design and reliable assessment, but all configurations achieved mediocre performance with high critical failure rates. Dataset augmentation decreased overall quality, suggesting methodological issues. The paper is well-written, organized, and transparent about limitations and failures. While the domain is important, the contributions are limited and the work mainly provides baseline metrics without advancing practical solutions. The focus is novel, but technical approaches are standard. Reproducibility is good, though use of commercial APIs may limit exact replication. Ethical considerations are thoroughly discussed. Related work coverage is adequate but could be more comprehensive. Major concerns include limited practical impact, insufficient evaluation scale, high failure rates, and the need for deeper investigation into context integration. Strengths include addressing an important challenge, rigorous methodology, transparent reporting, and strong ethical focus. Overall, the paper provides valuable negative results but its limited scope and lack of clear advancement constrain its impact.

---

### Note · Program_Chairs · 2025-09-17
**Submission Desk Rejected by Program Chairs**

Paper does not respect the conference requirements (e.g., Checklists and Formatting issues)

---

### Note · Reviewer_AIRevCorrectness · 2025-10-26

**Correctness Check**

### Key Issues Identified:

- Inconsistent reporting of key metrics across sections (e.g., baseline overall quality 3.14 vs 3.59; best configuration overall quality 3.39 vs 3.87; cultural relevance means reported as ~3.5 vs ~4.3–4.8). See Table 2 on page 3 vs Abstract and Section 4.2 vs Figure 1 caption on page 4.
- Conflicting sample size statements: n=100 in Table 2 and Figure 1 caption (page 4) versus "40 meal plans" in Section 4.2 and "100 total profiles" in the checklist (page 12).
- Mislabeling standard deviation as variance ("High variance (=0.98–1.39)") and unclear whether dispersion pertains to profiles or all evaluations.
- Lack of inter-rater reliability metrics (e.g., ICC/Krippendorff’s alpha), no rater blinding or training details, and unclear rater composition, undermining validity of MOS evaluations.
- No statistical significance testing or confidence intervals; only descriptive means ± SD and percent changes despite large variability.
- Under-specified implementation details for E2 (how the 79-food database was integrated into prompts) and E3 (query formulation, filtering, and incorporation of Exa API results).
- Stochastic generation at temperature=0.7 without multiple runs or seed control; no quantification of sampling variance.
- Ambiguity and late clarification of the 'critical failure' definition; reliance on overall quality ≤ 2/5 should be stated clearly up front.
- Incorrect or missing cross-references (e.g., references to Section 4.4 and Table 3 that do not appear in the provided text).
- Potential confounding from real-time web retrieval (temporal variability) without controls, affecting reproducibility.

---

### Note · Reviewer_AIRevRelatedWork · 2025-10-26

**Related Work Check**

Please look at your references to confirm they are good.

**Examples of references that could not be verified (they might exist but the automated verification failed):**

- Exa api: Neural search engine for academic and web content. by Exa Technologies

---

### Decision · Program_Chairs · 2025-10-08

**Decision:**

Reject

**Comment:**

Thank you for submitting to Agents4Science 2025! We regret to inform you that your submission has not been accepted. Please see the reviews below for more information.